# Molecular weaving via surface-templated epitaxy of crystalline coordination networks

Zhengbang Wang[1,*], Alfred Błaszczyk[2,3,*], Olaf Fuhr[2], Stefan Heissler[1], Christof Wöll[1] & Marcel Mayor[2,4,5]

One of the dream reactions in polymer chemistry is the bottom-up, self-assembled synthesis of polymer fabrics, with interwoven, one-dimensional fibres of monomolecular thickness forming planar pieces of textiles. We have made a major step towards realizing this goal by assembling sophisticated, quadritopic linkers into surface-mounted metal-organic frameworks. By sandwiching these quadritopic linkers between sacrificial metal-organic framework thin films, we obtained multi-heteroepitaxial, crystalline systems. In a next step, Glaser–Hay coupling of triple bonds in the quadritopic linkers yields linear, interwoven polymer chains. X-ray diffraction studies revealed that this topochemical reaction leaves the MOF backbone completely intact. After removing the metal ions, the textile sheets can be transferred onto different supports and imaged using scanning electron microscopy and atomic-force microscopy. The individual polymer strands forming the two-dimensional textiles have lengths on the order of 200 nm, as evidenced by atomic-force microscopy images recorded from the disassembled textiles.

[1] Institute of Functional Interfaces, Karlsruhe Institute of Technology, Hermann-von-Helmholtz-Platz 1, 76344 Eggenstein-Leopoldshafen, Germany. [2] Institute for Nanotechnology, Karlsruhe Institute of Technology, Hermann-von-Helmholtz-Platz 1, 76344 Eggenstein-Leopoldshafen, Germany. [3] Department of Commodity Science, Poznan University of Economics & Business, Aleja Niepodleglosci 10, 61-875 Poznan, Poland. [4] Department of Chemistry, University of Basel, St Johanns-Ring 19, 4056 Basel, Switzerland. [5] Lehn Institute of Functional Materials (LIFM), Sun Yat-Sen University (SYSU), 510275 Guangzhou, China. * These authors contributed equally to this work. Correspondence and requests for materials should be addressed to C.W. (email: christof.woell@kit.edu) or to M.M. (email: marcel.mayor@unibas.ch).

The fabrication of textiles, one of the most important forms of materials governing everyday life, is typically done in a top–down approach. The ability to use physical or mechanical methods to weave one-dimensional fibres (wool, cotton, polymers and so on.) with micrometre diameters into fabrics was an important advance in the history of mankind and still is an important element of present day technology[1–3]. Achievement of interwoven structures by self-assembly, that is, realizing a bottom-up instead of a top–down approach, remains a challenge. Certainly, the synthesis of two-dimensional (2D)-polymers[4] has been realized in a number of cases. Using rather harsh conditions, graphene sheets have been fabricated by temperature-induced decomposition of methane[5] or other hydrocarbons on metal surfaces[6]. Also, using much milder conditions, the polymerization of monolayers at the air–water interface has been reported[7]. However, textiles are fundamentally different from 2D-polymers. Whereas at the crossing points of 2D-polymers, different fibres are covalently interlinked, in textiles the fibre subunits are only connected mechanically and touch each other via weak intermolecular forces. Consequently, textiles are much more flexible than the corresponding 2D-materials since fibres can slide relative to each other and thereby distribute forces more easily.

As pointed out by Schlüter and coworkers[8], the rational synthesis of interwoven fibres from molecular subunits to yield textiles is a major challenge. Since nanotextiles consisting of interwoven, single-chain polymers are an interesting, novel form of textiles, there is great interest in providing synthetic approaches, particularly ones based on self-assembly of molecular subunits. While the induction of coupling reactions between appropriately functionalized linkers integrated into metal-organic frameworks (MOFs)[9] is a promising approach, the realization of flat, 2D textile sheets has not yet been demonstrated.

In this paper, we demonstrate an innovative approach to the fabrication of planar fabric sheets by employing programmed layer-by-layer (lbl) assembly of crystalline coordination networks using liquid-phase epitaxy[10]. The preoriented monomers were then converted by Glaser–Hay coupling into interwoven polymer threads, yielding true 2D textiles.

## Results

**Organic linkers synthesis and MOFs preparation.** While the standard solvothermal MOF or crystalline coordination network synthesis methods yield particulate powders[11], the lbl-scheme yields monolithic, crystalline and oriented MOF thin films[12]. In addition, this approach allows the manufacturing of sandwich-structures, where an active layer can be embedded between two sacrificial layers (Fig. 1a). The monomers, which eventually polymerize to form the target fabric, are perfectly preoriented in the MOF by attachment of suitable node anchoring groups (NAGs). The coupling of the NAGs to suitable metal-containing nodes is the basis for forming the crystalline MOF materials. The prearrangement of these 4t- monomers in the MOF so that their cross-linking reaction yields flat sheets is a crucial process to attain the goals of this project (Fig. 1b). We achieved the formation of true planar fabrics by separating the fibre-forming sheets within the MOFs by sacrificial layers (Fig. 1a). While such a prearrangement of active and sacrificial layers is difficult to achieve with conventional solvothermal MOF synthesis methods, our SURMOF-process affords in a straightforward fashion this type of multi-heteroepitaxy, that is, different types of MOFs stacked to obtain crystalline, oriented heterolayers[13]. The choice of a suitable organic linker to build up the molecular framework is crucial. The quadritopic (4t) units must contain at least two NAGs to yield the connectivity within the MOFs, and furthermore, two units are required for the polymerization yielding the polymer chains. When the lattice-constant of the MOFs matches the required distance between the two units for the polymerization, induction of the polymerization reaction by a suitable catalyst yields first the formation of a 2D fabric. A true interweaving of fibres will be realized by a statistical distribution of the north-south (NS) and east-west (EW) crossing points (Fig. 1b). In the absence of further external constraints, there will be an equal number of crossing points where the NS-connection

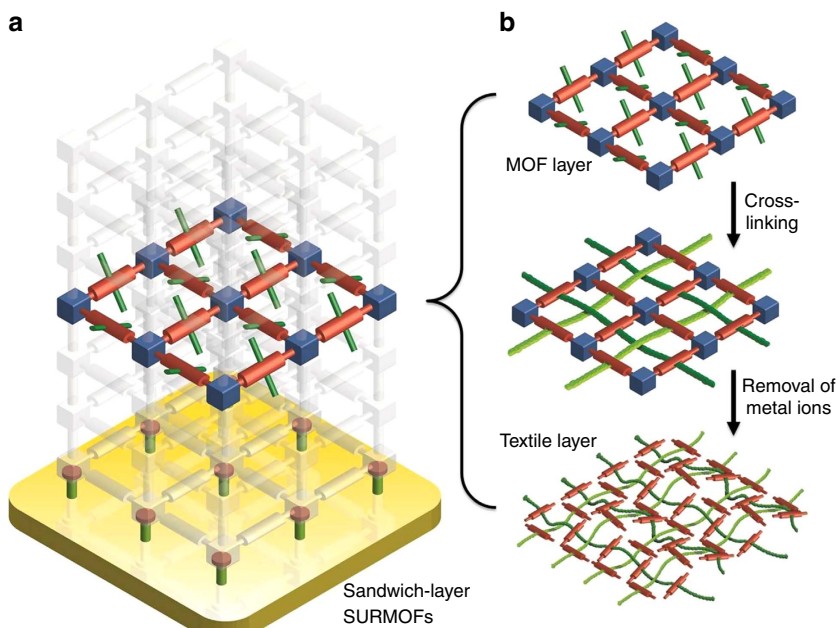

**Figure 1 | Strategy for the formation of molecular textile.** Schematic illustration of the heteroepitaxial sandwich-layer surface-mounted MOF (SURMOF) system (**a**) and the formation procedure of molecular weaving in the active MOF layer embedded between two sacrificial layers (**b**).

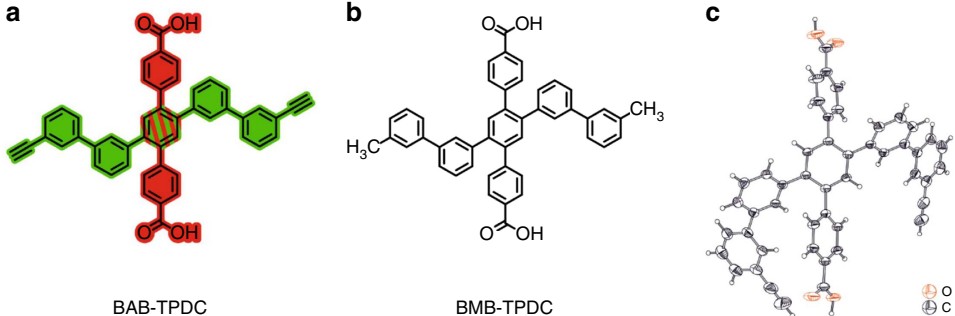

**Figure 2 | Structure of organic linkers. (a)** Chemical structure of quadritopic (4t) organic linker BAB-TPDC (bis(acetylene-biphenyl)-terphenyl di carboxylic acid). **(b)** Chemical structure of the ditopic linker BMB-TPDC (bis(methyl-biphenyl)-terphenyl di carboxylic acid). **(c)** Solid state structure of BAB-TPDC determined by X-ray crystallography (ellipsoids plotted at a 50% probability level).

is above or below the EW-connection and their distribution will be random. If the polymerizable group interconnecting two subunits of opposite edges results in a longer linkage than the distance of both 4t units in the MOF, the additional material must bend above or below the MOF plane to compensate for the size mismatch. If in a particular MOF cell the (NS) connection sidesteps above the plane, the (EW) connection must sidestep below the plane for steric reasons and vice versa. Additionally, if the polymerizable groups are rigidly fixed perpendicular to the revolving axis defined by the terminal NAGs, a sidestep below the MOF plane in a particular MOF cell forces a side step above the MOF cell in the neighbouring cell along the polymer direction and vice versa. In other words, the revolving character of the polymerizable subunits along the NAG-axis communicates the up/down information between neighbouring cells along a particular polymer chain. A suitably designed molecular building block should be able to profit from both effects to increase the extent of perfectly interwoven polymer strands. A fabric consisting of 1-d polymer chains without any strong ionic or chemical bonds between the chains is then achieved by removal of the metal ions by a complexation reaction.

The 4t unit (Fig. 2a), BAB-TPDC (bis(acetylene-biphenyl)-terphenyl di carboxylic acid), is a cruciform structure consisting of *para*-terphenyl and *meta*-quinquephenyl branches, with the central phenyl ring serving as a shared subunit. The rigid terphenyl rod is functionalized with terminal carboxylic acids in the 4 and 4″ positions (the NAGs). Carboxylic acids are perfectly suited as NAGs in the SURMOF-process, and similar approaches have been used very successfully for layer-pillar MOFs[14], where 2D-layers consisting of Cu(II)-carboxylate units (also referred to as paddle-wheels) are stacked. The single bonds interlinking the phenyl rings in the terphenyl backbone provide the required revolving freedom for the central phenyl ring, as well as for the perpendicularly arranged *meta*-quinquephenyl branch, which exposes alkynes in *meta*-positions as polymerizable groups. The quinquephenyl group was chosen deliberately. On one hand, its dimension forces the material out of the plane of the MOF structure while on the other hand, the *meta*-connections of the phenyl subunits provide the required degree of freedom enabling the polymerizable groups to explore the volume above and below the plane defined by the MOF-building terphenyls of a given layer.

The rationale behind choosing terminal alkynes as polymerizable groups is twofold. First, oxidative acetylene coupling using Glaser–Hay conditions[15,16] allows for the covalent interlinking of the subunits with the same exposed functional group, thus considerably reducing any symmetry issues of the monomer arrangement. Second, the reaction conditions applied are not only rather mild, but also require exclusively small reagents like copper ions and molecular oxygen, which are likely to be readily available

also in the spatially restricted pores of a MOF. To construct sacrificial layers which do not polymerize but otherwise have comparable spatial requirements and physical chemical properties, the ditopic linker, BMB-TPDC (bis(methyl-biphenyl)-terphenyl di carboxylic acid) (Fig. 2b, bis(methyl-biphenyl)-terPhenyl di carboxylic acid) was synthesized, which has both alkynes substituted by unreactive methyl groups. The assembly of both MOF subunits (BAB-TPDC and BMB-TPDC) is mainly based on Pd-catalyzed Suzuki coupling steps[17] and their syntheses and characterization data are provided in the Supplementary Method. To corroborate the identity of BAB-TPDC, single-crystal X-ray diffraction provided the solid state structures displayed in Fig. 2c. For the coupling of the 2D-paddle-wheel-planes, a standard and commercial pillar, dabco (1,4-diazabicyclo[2.2.2]octane), was used.

The X-ray diffraction (XRD)-data recorded for SURMOFs made from different linkers are provided in the supporting information (Supplementary Figs 1 and 2) and demonstrate that the BAB-TPDC and the BMB-TPDC linkers are well suited for the growth of highly oriented, crystalline layer-pillar-type SURMOFs. In addition, the data shown in Supplementary Figs 3 and 4 also demonstrate that these two linkers are well suited for the multi-heteroepitaxy, that is, BAB-TPDC SURMOFs can be grown on top of BMB-TPDC SURMOFs and vice versa. Furthermore, the alternate growth of one cycle of BAB-TPDC SURMOFs and one cycle of BMB-TPDC SURMOFs (called ABAB-SURMOFs, where A is one cycle of BAB-TPDC SURMOFs and B is one cycle of BMB-TPDC SURMOFs) was also carried out successfully using a modified liquid-phase epitaxy method, as shown in Supplementary Figs 5 and 6. The XRD pattern recorded for the SURMOF bilayer shows no substantial difference to the corresponding homogenous SURMOFs made from either organic linker A or B (see simulated results in Supplementary Fig. 7). An important issue with regard to achieve embedded (between sacrificial layers) polymer precursors is the possibility of linker exchange occurring when a MOF built with linker A is immersed in a solution containing linker B (refs 18,19). To demonstrate that this effect can be neglected in our case, we have carried out control experiments where quantitative infrared (IR) spectroscopy was used to measure the amount of replaced linkers. As shown in Supplementary Fig. 8, the vibrational mode at 3,295 cm$^{-1}$ characteristic of the C–C triple bond in the BAB-TPDC linker grown on the top layer of the SURMOFs (Supplementary Fig. 8a) increased quickly while the new top-layer was formed, but on continuous soaking of the sample in the solution to favour linker exchange with the layer below for 24 h, only a very minor further increase (3.1%) was observed. A similar observation was made for the opposite situation, for details see Supplementary Fig. 9. Here, the change

between immersion for 30 min and 24 h amounted to only 6.8% of 24 h. These very low rates for linker exchange can be rationalized by the rather bulky characters of both linkers. In this context, it is noteworthy that initial attempts using *p*ara-terphenyl 4,4″-dicarboxylic acid as linker in the sacrificial layer were not successful due to substantial linker exchange. Only when using the more bulky sacrificial linker BMB-TPDC high-quality mixed-layer SURMOFs could be fabricated. In previous cases reported in the literature where larger exchange rates were observed the linkers were considerable less bulky[18,19].

### Formation of molecular threads.

The Glaser–Hay coupling of the acetylene side groups of the BAB-TPDC linkers was induced by a catalyst, a tetramethylethylenediamine (TMEDA) complex of copper(I) chloride (Cu-TMEDA)[15] (Fig. 3a). The consumption of the C–C triple bond can be readily determined by monitoring the intensity decrease (Fig. 3b) of the characteristic IR vibrational band at $3,295\,cm^{-1}$. A quantitative analysis of the peak areas indicated a conversion rate of 80%. Raman spectroscopy revealed the corresponding formation of the diacetlyene coupling product (band at $2,216\,cm^{-1}$, Fig. 3c). We attributed the presence of a weak band at this position before the reaction to a small fraction of diacetlyene units formed during the MOF synthesis. The other characteristic vibrational bands in the IR and Raman data showed no variation on the coupling reactions, suggesting that the SURMOF backbone remained unchanged in the course of the Cu-TMEDA-catalyzed coupling reaction. This conclusion is supported by the XRD-data recorded after the conversion (Fig. 3d), which demonstrate that the framework kept its crystallinity during the coupling reaction. The slight decreases of the diffraction peak intensities and a small shift of peak positions towards higher angles can be related to the slight strain as a result of the coupling reaction between adjacent linkers. This hypothesis is corroborated by the structure simulation of the MOFs before and after the

coupling reaction (Supplementary Fig. 10). Note that because of steric constraints, only 4t-units located opposite each other can be coupled via the Hay reaction; for other cases, the distances between the acetylene subunits are either too long or too short. In the next step, the metal ions were removed by soaking the reacted SURMOF sample in dilute hydrochloric acid solution, which was monitored by infrared reflection–absorption spectroscopy (IRRAS). The occurrence of vibrational bands characteristic for carboxylic acids are evident in the IR spectrum after acid treatment[20] (Fig. 3b, C=O-vibrations in isolated acids groups at $1,730\,cm^{-1}$ and in dimeric acid groups at $1,700\,cm^{-1}$).

The nature of the thin layers of interwoven polymer chains resulting from the Glaser–Hay coupling and subsequent removal of the metal ions could be studied more conveniently by scanning electron microscopy (SEM) and atomic-force microscopy (AFM). To allow for such a characterization, the molecular textiles were transferred from the templating SURMOF substrates to either TEM grids or reference substrates using a lift-off technique (Supplementary Fig. 11)[21,22] SEM- and AFM-images of the reaction product clearly showed the presence of planar textiles, see Fig. 4a,b. Using AFM, the thickness of the polymer transformed to a smooth Si substrate was determined and found to be ∼20 nm (Fig. 4c), which demonstrated the presence of the multilayers of molecular weaving. Note that to obtain sufficient intensity in the IR and Raman spectra, in this case 10 cycles of ABAB-SURMOFs described above were used. An ABAB sequence was used to avoid the unwanted reaction between the 4t-linkers of different A layers. The acetylene side groups are too short to react with the next A layer through a separating B layer. In addition, the large, non-activated side groups occupied the space (Supplementary Fig. 12), which hindered the crossing of active side groups though the sacrificial layer.

To obtain a true planar, single sheet of fabric, we decreased the number of active layers in the sandwiched ABAB SURMOF. As shown in Fig. 4d,e, textiles prepared from 5 cycles of

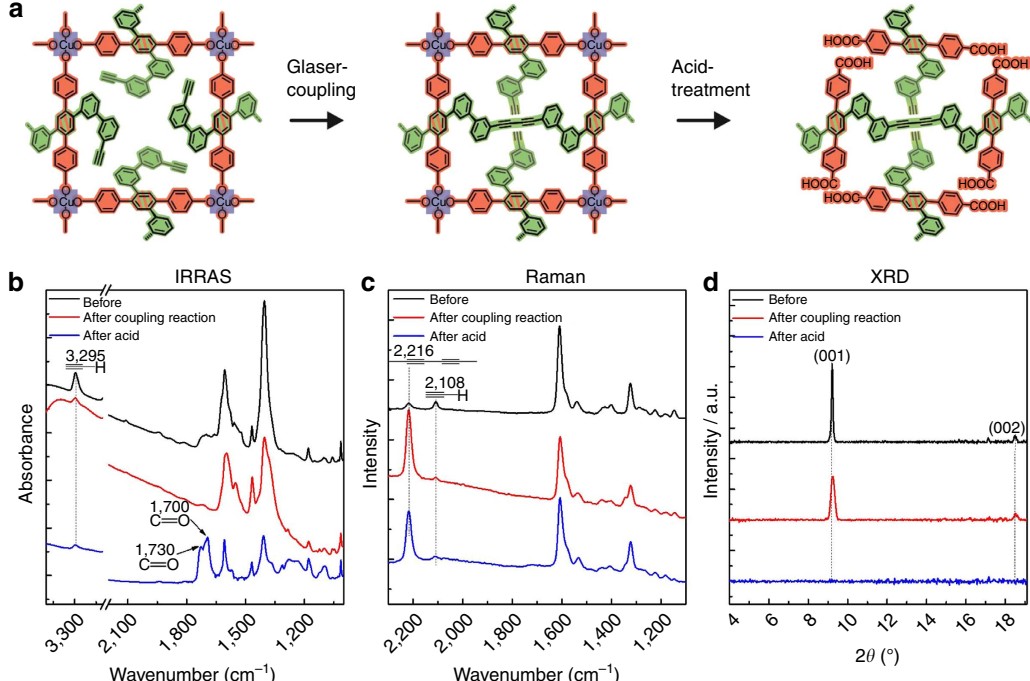

**Figure 3 | Formation of molecular textiles.** (**a**) Schematic illustration of the formation of molecular weaving in the active MOF layer constructed by BAB-TPDC quadritopic (4t) organic linker. (**b**) IRRAS, (**c**) Raman and (**d**) XRD results collected from the samples with 10 cycles ABAB-SURMOF system with a repeated growth of one layer of BAB-TPDC SURMOFs (MOF-A) and one layer of BMB-TPDC SURMOFs (MOF-B), here one cycle includes one layer of A and one layer of B. In **b**–**d**, the initial spectrum is black, the spectrum after coupling is red, and spectrum after acid treatment is blue.

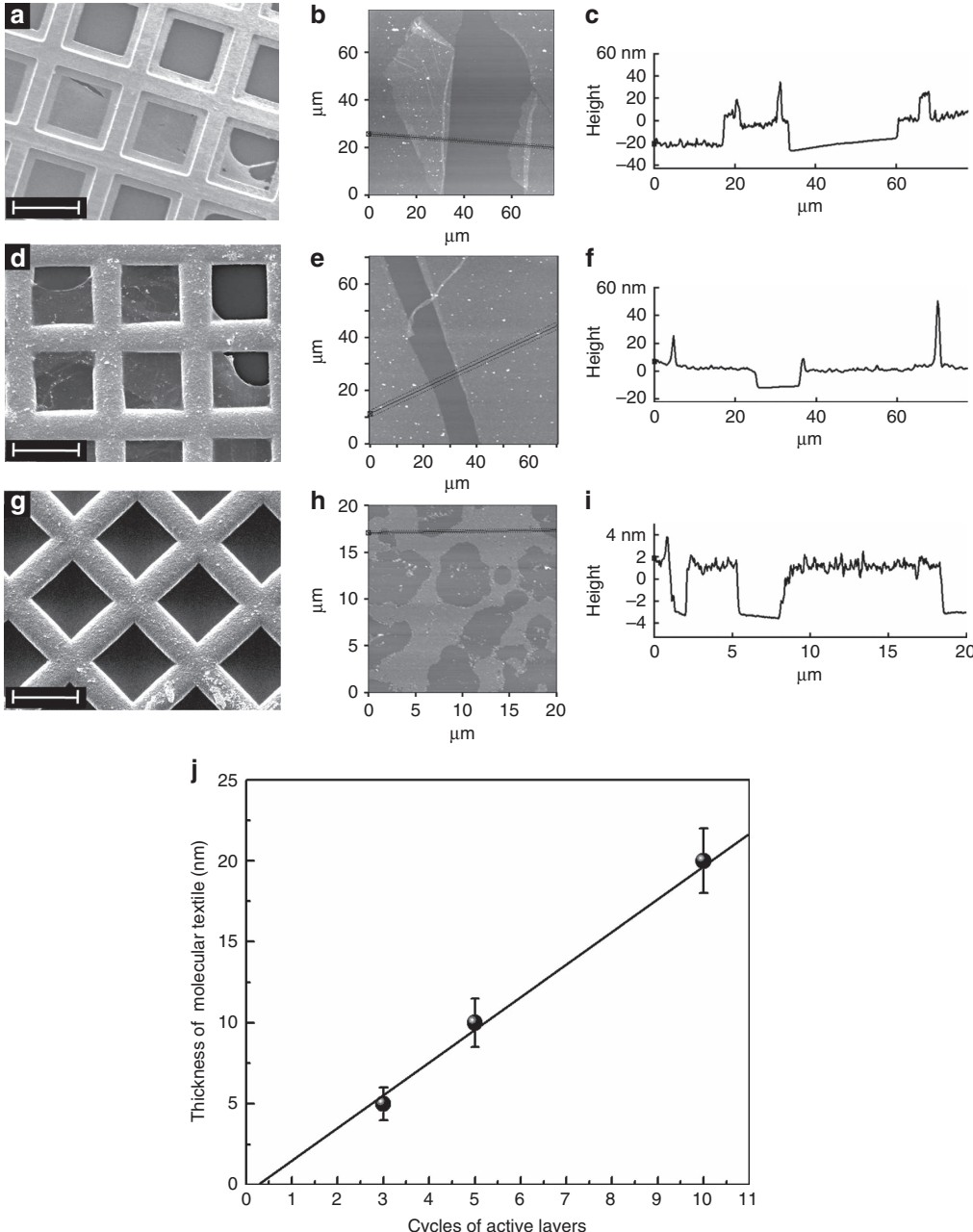

**Figure 4 | Imaging of molecular textiles.** (**a**) SEM image of 10 cycles of molecular weaving attached on TEM grids. (**b**) AFM image and (**c**) height profile of 10 cycles of molecular weaving attached on a smooth Si substrate. (**d**) SEM image of 5 cycles of molecular weaving attached on TEM grids. (**e**) AFM image and (**f**) height profile of 5 cycles of molecular weaving attached on a smooth Si substrate. (**g**) SEM image of 3 cycles of molecule weaving attached on TEM grids. (**h**) AFM image and (**i**) height profile of 3 cycles of molecular weaving attached on a smooth Si substrate. (**j**) Dependence of the textiles thickness, as measured by AFM, on the number of cycles to form ABAB-SURMOFs active layers. The length of the white scale bar in the SEM images is 50 μm.

ABAB-SURMOFs could be still clearly observed in the SEM images, and the thickness was ∼10 nm (Fig. 4f). Unfortunately, when the active layers were decreased to 3 cycles, we could not observe a continuous fabric in the SEM images (Fig. 4g); however, small broken patches of the thin fibre films were found in the AFM image (Fig. 4h) with a thickness of ∼5 nm (Fig. 4i). We attribute the fact that the pieces become smaller when decreasing the number of cycles to both, the roughness of the SURMOF thin film and the number of coupling failures between neighbouring acetylenes, probably due to mismatches in the spatial arrangement above and below the plane defined by the SURMOF layer. We expect that further optimization of SURMOF formation

conditions (for example, the use of linker oligomers) will allow for the further increase in the size of the textile patches. In this context, it is also important to note that the SURMOF-process can also be realized using a spray process[23], which, in principle, allows a continuous fabrication process allowing the manufacture of these 2D-textiles on a larger scale.

**Disassembly of the 2D fabrics into individual polymer strands.** As indicated in Fig. 5a, more information on the individuals polymer strands forming the 2D textiles can be obtained after disassembling the obtained fabric using ultrasonication in THF

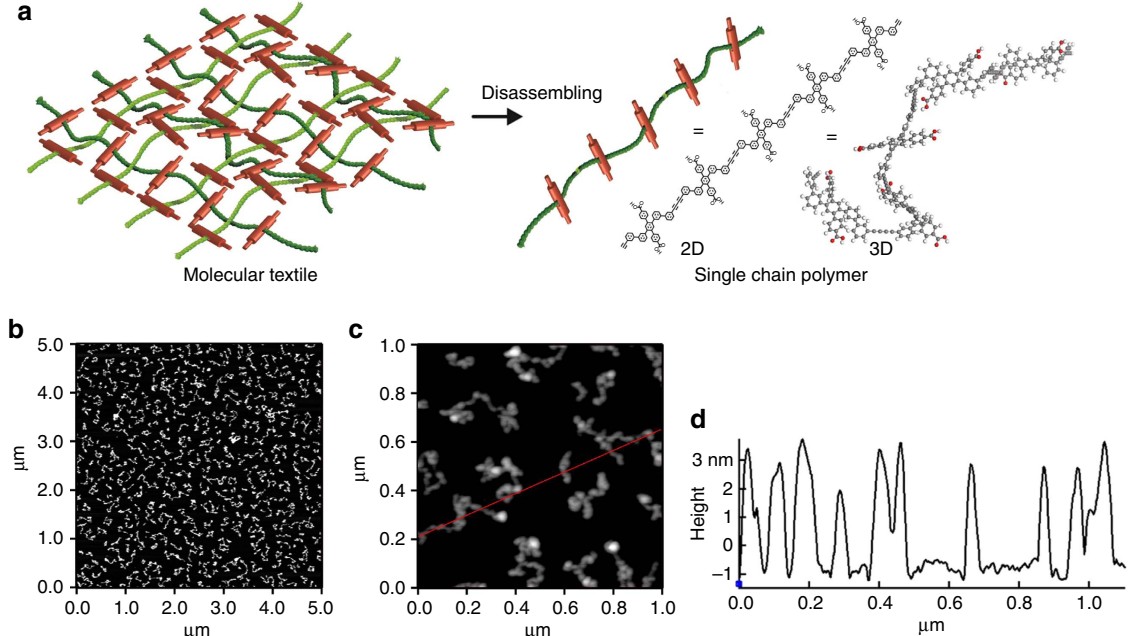

**Figure 5 | Disassembly of molecular weaving to single-chain polymers. (a)** Schematic illustration of the disassembling of molecular weaving to single-chain polymers. **(b,c)** AFM images of the single-chain polymer prepared by disassembling the molecular weaving in tetrahydrofuran solvent under ultrasonic condition and the AFM sample was prepared by drop-casting the diluted solution on smooth Si substrate. **(d)** Height profile of the single-chain polymer along the red line inside the AFM image (**c**).

(for details see the SI) for 6 h. After drop-casting the diluted solution on a smooth Si substrate, the single strands could be unambiguously identified in the corresponding AFM images, see Fig. 5b,c. The individual polymer chains exhibit a heights of around 4 nm (Fig. 5d), which is slightly larger than the molecular dimensions (1.6 nm) of the polymer precursors (Supplementary Fig. 13). However, considering the *meta*-oligophenyl backbone of the polymer strand allowing to swell by coiling in a spring like arrangement as shown in Figs 3d and 5a, the observed dimensions match perfectly the expectations for the polymer strands. The thickness of $\sim 5$ nm observed for the fabric before disassembly (Fig. 4h) together with the thickness of the individual polymer chains ($\sim 4$ nm, see above) suggests that the fabricated textile sheets consist of at most double layers. The length of the polymer threads forming the 2D fabric can be estimated from the AFM images and amounts to $\sim 200$ nm. This value is slightly larger than the average lateral size of the SURMOF domains ($\sim 80$ nm, as deduced from the width of the in-plane XRD (100) peaks, see Supplementary Fig. 6) and might even point at interlinking of not yet reacted neighbouring parallel polymer strands at the edge of polymer patches.

## Discussion

In conclusion, we present an innovative strategy to produce 2D-polymers consisting of interwoven linear polymer fibres which are held together exclusively by the mechanical forces arising from the weaving pattern. The linear polymer fibres, warp and weft, are formed by oxidative acetylene coupling between monomers which are perfectly pre-organized in SURMOF-layers. While the analytical data and physical properties corroborate the claimed supramolecular network, it also demonstrates the current limitations in accessible dimensions of molecular 2D-textiles.

Currently we are working on further exploring the potential of SURMOFs as organizers of molecular subunits. In particular, we seek alternative linking chemistry, weaving patterns and stimuli responsive functionalities.

## Methods

**X-ray diffraction (XRD).** XRD measurements for out-of-plane (co-planar orientation) were carried out using Bruker D8-Advance diffractometer equipped with a position sensitive detector Lynxeye in $\theta - \theta$ geometry, variable divergence slit and 2.3° Soller-slit was used on the secondary side. XRD measurements for in-plane (non-co-planar orientation) were carried out using Bruker D8 Discover equipped with a quarter Eulerian cradle, tilt-stage and 2.3° Soller-slits were installed in both sides. A Göbel-mirror, and a position sensitive detector Lynxeye in $\theta - 2\theta$ geometry, was applied in the measurement. The Cu-anodes which utilize the Cu $K_{\alpha 1, 2}$-radiation ($\lambda = 0.154018$ nm) was used in both instrument. The measurement was carried out in the range of $2\theta = 4° - 20°$ at a scan step of 0.02° at 40 kV and 40 mA. The samples using for the XRD measurement were grown on gold substrate (see SURMOF preparation section) with 40 growth cycles with a thickness of around 40 nm.

**Infrared reflection–absorption spectroscopy (IRRAS).** The IRRAS of the samples in this study were acquired with a resolution of $2 \, \text{cm}^{-1}$ using a FTIR spectrometer (Bruker VERTEX 80v). All the IRRAS results were recorded in grazing incidence reflection mode at an angle of incidence amounting to 80° relative to the surface normal using liquid nitrogen cooled mercury cadmium telluride narrow band detectors. Perdeuterated hexadecanethiol self-assembled monolayers (SAMs) on Au/Si were used for reference measurements.

**Raman.** The Raman-spectra were recorded with a Bruker Senterra Raman microscope (Bruker Optics, Ettlingen, Germany) using a green laser at 532 nm for excitation. The integration time was 60 s with 3 co-additions.

**Scanning electron microscope (SEM).** The SEM measurements were carried out using a Philips XL SERIES 30 ESEM-FEG. To increase the image contrast and the resolution, the samples were coated with a 5 nm thick gold layer before recording the SEM micrographs.

**Atomic force microscopy (AFM).** AFM-imaging was done using an Asylum Research Atomic Force Microscope, MFP-3D BIO. The AFM was operated at 25 °C in an isolated chamber in alternating current mode (AC mode). AFM cantilevers were purchased from Ultrasharp MikroMasch. Three types of AFM cantilevers were used, an NSC-35 (resonance frequency 315 kHz; spring constant 14 N/m), an

NSC-36 (resonance frequency: 105 KHz; spring constant: 0.95 N/m) and an NSC-18 (resonance frequency: 75 kHz; spring constant: 3.5 N/m).

**Nuclear magnetic resonance (NMR) spectroscopy.** [1]H-NMR and [13]C-NMR spectra were recorded on a Bruker NMR 500 Instrument, the $J$ values are given in Hz. The solvent and temperature of each investigation is given at the start of the array listing the recorded signals.

**High-resolution mass spectrometry (MS and HRMS).** MALDI-TOF spectra were recorded using a Waters Synapt time-of-flight mass spectrometer. The electron ionization mass spectrometry (EI-MS) measurements were recorded using a LKB-9000S instrument.

**Single-crystal X-ray diffraction.** Single crystals of $C_{56}H_{46}O_6$ (BAB-TPDC) were obtained by slow diffusion of hexane into a solution of BAB-TPDC in THF. A suitable crystal was selected and mounted on a 'STOE IPDS2' diffractometer. The crystal was kept at 180.15 K during data collection. Using Olex2 (ref. 24), the structure was solved with the ShelXS (ref. 25) structure solution programme using direct methods and refined with the ShelXL (ref. 26) refinement package using least squares minimization.

**Self-assemble monolayer (SAM) preparation.** For SAM formation a clean gold substrate (gold coated silicon or gold coated mica) was rinsed with pure ethanol and then immersed in a solution of (4-(4-pyridyl)phenyl)methanethiol (PP1) with a concentration of $1 \text{ mmol l}^{-1}$ in ethanol for 18 h. Afterwards the substrate was taken out, rinsed thoroughly with ethanol and dried under nitrogen stream.

**SURMOF growth.** The experimental procedure used to grow MOFs on the organic surface has been discussed in detail previously[10,27]. In short, the epitaxial growth process consisted of alternately immersing a (4-(4-pyridyl)phenyl)methanethiol (PP1) SAMs modified substrate into ethanolic solutions of the building units: copper acetate (1 mM) and BAB-TPDC (or BMB-TPDC)/dabco (equimolar ratio of 0.05 mM) organic linkers ethanolic solution. Between each immersion step, the substrates are rinsed thoroughly with ethanol. In the present work, the SAM substrates were immersed into copper acetate solution for 15 min, subsequently rinsed with pure ethanol solution for 2 min, and then immersed into linker solutions for 30 min. All the solutions were kept at 60 °C during MOF film preparation. Note, during the synthesis process, the organic liners can be changed to produce the MOF-on-MOF (ref. 13) or ABAB-SURMOFs systems (Supplementary Fig. 5).

**Glaser-coupling reaction.** Freshly prepared samples of SURMOFs were immersed in a solution of dichloromethane containing 0.05 mM of TMEDA complex of copper(I) chloride (Cu-TMEDA) to process the cross-linking reaction for 72 h under $O_2$ atmosphere. Afterwards the samples were thoroughly rinsed with dichloromethane and ethanol and dried under nitrogen flow.

**Acid treatment to receive molecular textiles.** Cross-linked SURMOF samples were immersed in 1 mM dilute hydrochloric acid solution and the solution was refreshed for three times every 30 min. Afterwards the samples were rinsed thoroughly with ethanol and water and dried under nitrogen flow.

**Transformation of molecular textiles to other supports.** In this case, the SURMOFs were grown on PP1 modified gold coated mica substrate. After Glaser-coupling reaction and acid treatment, PMMA was spin coated as a supporting layer and afterwards the mica was removed by immersing in $I_2/KI/H_2O$; $KI/H_2O$ and in the last step $H_2O$. The retaining gold film was etched in a solution of $I_2/KI/H_2O$. The membrane was washed using water for three times. Afterwards the membrane was transferred to the desired substrate. The PMMA layer can be dissolved by acetone. Note, the process was also displayed in Supplementary Fig. 11.

**Disassembling of molecular textiles to single-chain polymer.** First, the molecular textile film was immersing in THF solvent and the disassembling was performed via a physical dispersion of the molecular weaving under ultrasonic condition for 6 h. The diluted solution was drop-casted on smooth Si substrate for AFM measurement.

**Synthesis of organic linkers BAB-TPDC and BMB-TPDC.** The synthesis of organic linkers BAB-TPDC and BMB-TPDC is described in Supplementary Fig. 14.

**Data availability.** CCDC-1489009 contains the supplementary crystallographic data for this paper. These data can be obtained free of charge via www.ccdc.cam.ac.uk/conts/retrieving.html (or from Cambridge Crystallographic Data Centre, 12 Union Road, Cambridge CB21EZ, UK; fax: (+44)1223-336-033; or deposit@ccdc.cam.ac.uk). The data sets generated during and/or analysed during the current study are available from the corresponding author on reasonable request.

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

## Acknowledgements

We gratefully acknowledge funding for this project from the Sonderforschungsbereich SFB 1176 (project C6) of the German Research Council (DFG), and by the Swiss National Science Foundation (SNF, 200020-159730).

## Author contributions

All authors contributed to writing the manuscript and have approved the final version of the manuscript. Z.W. synthesized the samples and performed the XRD, IRRAS, SEM and AFM characterizations. A.B. synthesized the organic linkers and performed the NMR and MS measurement. O.F. performed the single-crystal X-ray diffraction measurement. S.H. performed the Raman measurement. C.W. and M.M. supervised the experiments.

**Additional information**

**Competing financial interests:** The authors declare no competing financial interests.

