## [Peer Review File · Nature Communications]

Reviewers' Comments:

Reviewer #1 (Remarks to the Author)

In this contribution the author presented a very smart way to do topochemical reaction in MOFs. By using a surface-mounted MOF, the author introduced directionality into the framework, by which an ordered multivariate MOF can be formed and the topochemical reaction of linkers within one plane would proceed without interference with reactive functional groups of linkers from other layers. By employing this topochemical reaction, the author claims that they made a 2-D interwoven framework which is usually difficult to achieve and unprecedented in the literature. We found this strategy to exclusively made 'in-plane bonds' is powerful and unique, which makes this paper very intriguing, however, this technical part of this paper could potentially be improved.

1. The characterization of the surMOF. The author did the routine characterization for the MOF, but there are two outstanding issues need to be addressed. The first one is the ratio between the alkyne functionalized and un-functionalized linker should be 1:1. We believe this ratio could be easily studied by digested-NMR and is a vital parameter in this case. Even with the correct ratio, one could still argue that during the layer by layer growth, linker exchange might also take place and in the sacrificial MOF layer some reactive linker might also be present (although epitaxial growth of two kind of MOF is known, but the precision of layer by layer growth the authors are arguing here has not been proved and is non-trivial for their application) . Moreover, if the structure of the surMOF is really as depicted by the authors with two type of layers being alternative with each other, a super-lattice peak corresponding to twice the spacing of the c direction (dabco direction) should be resolved in the out of plane XRD diffraction, which is not discussed in the paper. Without explicitly addressing this two issues, it is not so clear whether the expected structure has been made successfully. Even the real MOF just has slight deviation from the ideal proposed structure, the authors' claim for interwoven layer would be significantly compromised.

2. Even we assume that the above mentioned MOF is made in 100% precision, whether the interwoven layers were made is still questionable. Although the author shows that the coupling did took place, the completion of the reaction is not discussed. In the AFM the resolution is too low to resolve any detail information on the layered structure and the layer thickness. Essentially, there is no direct proof of the presence of the atomic thickness of this film at all. We believe that the characterization of the 'woven layer' is too preliminary. We suggest the authors to either revise their claim on the atomic thickness of the layer and the detailed molecular structure of the layer or perform high resolution AFM to see at least the molecular structural details within one exfoliated layer. At least some sub 50 nm feature should be resolved in terms of the thickness of the layers. In summary we found this report very interesting but we strongly suggest the authors to perform detailed characterization of their material before pursuing publication of this paper in nature communication.

(recommendation: not suitable for publication, resubmit or submit else where)

Reviewer #2 (Remarks to the Author)

I read this paper with joy. Attempts to create regular, large scale woven materials on a molecular

scale are a major challenge and definitely worth pursuing. Achieving such goal is associated with an extraordinarily high level of static and dynamic structural control. A molecular fabric represents a breakthrough in the molecular sciences combined with a promising step ahead in terms of dimensionally restricted, new materials with a great potential for applications. I thus consider the manuscript principally important.

Having said this, there are a number of deficiencies in the work presented, however, and it is thus too early for such claim to be made.

If the authors had performed the key experiment to prove a few layer or monolayer material to be a molecular fabric, evaluation of the manuscript would be easier. This experiment concerns the fact that fabrics can in principle disassemble into their constituents, which are the threads. In carpets this is prevented from happening on purpose by implementing physical knots or turning the threads after each weaving step. Because no such measures were taken by the authors, the edges of their sheets are critically important. They should suffer fraying-out phenomena that eventually result in complete disassembly. This disassembly should result in linear polymers which can be proven by conventional polymer analytical methods such as GPC.

Comparing the present manuscript with the recent publication by Yaghi in which he and his collaborators claim to have achieved 'molecular weaving' the significant step ahead is the fact that the present work provides sheet-like entities, while Yaghi's work affords 'only' microcrystalline powders. A material to choose for doing disassembly studies, is certainly a sheet and not a powder. If there happens to be kinetic hindrance, the disassembly could be enforced with AFM tips. There are useful experiments reported.

There are a number of issues with the manuscript some of which are delineated below.

1. What quantitative argument makes the authors believe that only the desired connections between the terminal acetylene units are formed? Given the rotational freedom of the meta-linked units – by gut feeling - other connections should be feasible as well. The possible built-up of ring strain associated with this is likely to be very small and in any case distributes over a large number of atoms
2. What makes the authors state that the acetylene conversion is approximately 90%. Looking at Figure 3b quantification is not an easy task but by visual inspection the present reviewer would estimate at a significantly lower value (possibly 60-70%; which is already an achievement). For a 'true' woven network, as the authors refer to their product, 100% conversion would be needed.
3. Another issue concerns the absence of monolayers. Obviously the authors can make layered MOFs in which sacrificial layers and active layers alternate. Why not using them as starting material for a molecular fabric? This should work if all the assumptions made actually apply.
4. On a side note: The strength of a fabric does not normally outperform a covalent network but rather its ductility. Graphene has an enormous strength.

The reviewer feels being in sort of a dilemma. On the one hand the idea is beautiful and work in this direction should be pursued by all means. On the other hand, there are a number of serious issues (some of which are mentioned) which do not convince the reviewer about the full correctness of the claims made. If this submission was a perspective paper, it could be possibly accepted. As full scale research paper though this cannot be the recommendation. The present reviewer would be very willing to reconsider if substantial improvements have been achieved.

Reviewer #3 (Remarks to the Author)

The authors of this work describe a compelling strategy to use layer-by-layer MOF crystals create inter-woven single-strand polymers across large dimensions, which if realized, would constitute an interesting advance in the scaling of polymer textiles. The idea presented involves multi-hetero-layering of pillared 2D MOFs using two sets of binodal linkers, one of which couples only to the CuII metal centers, and the other with active alkyne-terminated side groups that are specifically sized to enable polymer cross-linking across the pores in the 2D polymer plane. While the research concept and description are intriguing, the primary results, including TEM and AFM analysis, unfortunately do not conclusively show that the resulting material contains interwoven molecular units as shown and described in the diagrammed scheme. Therefore, I cannot recommend publishing this article.

Specific Comments:

1. The TEM and AFM results show that the thickness of the polymer layer left behind after removing the MOF metal-centers is directly proportional to the number of layers of the MOF thin film deposited by the sequential dip method. However, this is certainly not sufficient to conclude that the molecular weaving has been achieved. In fact, this trend could also result if the molecules were deposited in randomly oriented monolayers before polymerization. To support the authors' claim, the authors should at least do control experiments varying the ABAB sequence, for example, doing AABAAB and show that the scaling trend in Figure 4 still holds. Moreover, the authors need to describe what the expected layer thickness would be for the schematic shown. Figure 4 shows 2 nm per layer which is too large for the expected laterally aligned woven molecular bilayer.
2. Another concern is that for the mechanism to proceed as diagrammed, we must believe that the MOF layer forms as a uniform 2D monolayer across a large substrate surface, and that linker exchange between the layers does not occur. The similarity of the BAB-TPDC and BMB-TPDC linkers will facilitate exchange. Missing linkers are also known to often appear during MOF growth. For the process to create "perfectly interwoven polymer strands" and "perfectly pre-organized...SURMOF layers" (as the authors claim) we need to assume that the process produces no linker exchange and no missing linker sites. A missing linker in non-polymerizing layer "B" for example, could be filled by a polymerizing linker during layer "A" formation. This linker intermixing will preclude formation of "true planar fabrics", in direct opposition to the authors' key claim. A careful reading of these authors' previous work on MOF heterostructures (e.g. ref #13 in current manuscript) where each hetero-layer consisted of multiple layers of each MOF, does not convincingly exclude the possibility of linker exchange or missing linkers during growth.
3. It is well known that in many 2D material growth strategies, such as molecular beam epitaxy, system thermodynamics can drive formation of clusters and multilayers even under conditions limited to quantitative monolayer coverage. The results showing patchy pieces after 3 cycles, and the authors' acknowledgment of the appearance of surface roughness support the view that full 2D MOF growth is not achieved.
4. The authors also state that the MOF layering method yields "monolithic crystalline oriented MOF films." They do not say over what lateral dimension the crystals will extend. This limitation would also determine the limit for the lateral size of the resulting "textile". Perhaps if the authors were able to quantify the molecular weight of the resulting polymer they could compare it to the expected lateral dimension of the starting MOF crystal. It is important for the authors to describe what limits the lateral dimensions of the MOF crystals.
5. More information is needed related to the in-plane diffraction experiment. How thick were the films and how were they prepared for clean in-plane scattering? This information is not clearly described in the experimental section in the manuscript or SI.

Reply to the reports of the referees

We thank all three referees for a very careful reading of the manuscript. While all reviewers recognized the importance and novelty of our work, in fact all three raised concerns as regards the requirement of more information of the textiles, and in particular on the nature of the polymer chains. In following the suggestion of one of the referees, we have disassembled the textiles using ultrasonic treatment and have subsequently imaged the individual polymer strands with AFM. These new results beautifully show the polymer chains and, in a very convincing fashion, demonstrate that the reaction scheme provided in our manuscript actually correctly describe the polymerization reactions.

We are optimistic that this new data, together with the point-by-point rebuttal of the criticism of the referees, will allow the referees to come to a positive recommendation.

Certainly, the new experiments suggested by the referees have led to a substantial improvement of the manuscript.

Response to reviewers' comments:

Reviewer #1 (Remarks to the Author):

In this contribution the author presented a very smart way to do topochemical reaction in MOFs. By using a surface-mounted MOF, the author introduced directionality into the framework, by which an ordered multivariate MOF can be formed and the topochemical reaction of linkers within one plane would proceed without interference with reactive functional groups of linkers from other layers. By employing this topochemical reaction, the author claims that they made a 2-D interwoven framework which is usually difficult to achieve and unprecedented in the literature. We found this strategy to exclusively made 'in-plane bonds' is powerful and unique, which makes this paper very intriguing, however, this technical part of this paper could potentially be improved.

Response: We are very pleased to learn that this reviewer realized the importance of our paper. As discussed in detail below, we have carried out additional experiments and product characterizations, which lead to a substantial improvement of the manuscript.

1. The characterization of the surMOF. The author did the routine characterization for the MOF, but there are two outstanding issues need to be addressed. The first one is the ratio between the alkyne functionalized and un-functionalized linker should be 1:1. We believe this ratio could be easily studied by digested-NMR and is a vital parameter in this case. Even with the correct ratio, one could still argue that during the layer by layer growth, linker exchange might also take place and in the sacrificial MOF layer some reactive linker might also be present (although epitaxial growth of two kind of MOF is known, but the precision of layer by layer growth the authors are arguing here has not been proved and is non-trivial for their application). Moreover, if the structure of the surMOF is really as depicted by the authors with two type of layers being alternative with each other, a super-lattice peak corresponding to twice the spacing of the c direction (dabco direction) should be resolved in the out of plane XRD diffraction, which is not discussed in the paper. Without explicitly addressing this two issues, it is not so clear whether the expected structure has been made successfully. Even the real MOF just has slight deviation from the ideal proposed structure, the authors' claim for interwoven layer would be significantly compromised.

Response: We agree with the reviewer that a high precision is needed when – during the SURMOF growth process - changing from the unfunctionalized to the functionalized linker. The fact that the layer-by-layer method can be successfully used to realize such sharp interfaces has been demonstrated in previous work (Z. Wang et al., Nano Letters 2014, 14, 1526-1529). In order to demonstrate that there the linker exchange is negligible we have

carried out IR-experiments for ABA-systems (B being the functionalized linker). The total amount of B in a deposited SURMOF can be determined from the intensity of the C-C triple bond at 3295 cm^{-1} , see Fig. 10 b in the SI.

The distinct vibrational mode at 3295 cm^{-1} observed after soaking 20 cycles of $\text{Cu}_2(\text{BMB-TPDC})_2(\text{Dabco})$ SURMOFs in 0.05/0.05 mM BAB-TPDC/Dabco solution at $60\text{ }^\circ\text{C}$ for 30 min can be assigned to the C-C triple bond in the BAB-TPDC linker grown on the top layer of the SURMOFs (see Fig. S10a). When continuing to soak the sample in the solution for 24h, a further increase of the integrated band area of 3.1 % was observed. In the opposite case the increase amounted (in 24 h) to 6.8%, indicating that linker exchange during the typical immersion times used here is negligible. A corresponding remark has been added to the text of the revised manuscript.

The reviewer also suggested that the super-lattice structure formed in this layer-by-layer procedure should be observable for AABBAABBAABB structures via the occurrence of additional peaks in the out-of-plane XRD data. In principle, the referee is correct. Unfortunately, however, the intensities of such superlattice peaks are expected to be very weak, since the electron density difference between the two types of linkers is very small (methyl and alkyne). This consideration explains why, in the corresponding experiments, we failed to observe such additional superlattice spots.

2. Even we assume that the above mentioned MOF is made in 100% precision, whether the interwoven layers were made is still questionable. Although the author shows that the coupling did take place, the completion of the reaction is not discussed. In the AFM the resolution is too low to resolve any detail information on the layered structure and the layer thickness. Essentially, there is no direct proof of the presence of the atomic thickness of this film at all. We believe that the characterization of the 'woven layer' is too preliminary. We suggest the authors to either revise their claim on the atomic thickness of the layer and the detailed molecular structure of the layer or perform high resolution AFM to see at least the molecular structural details within one exfoliated layer. At least some sub 50 nm feature should be resolved in terms of the thickness of the layers.

In summary we found this report very interesting but we strongly suggest the authors to perform detailed characterization of their material before pursuing publication of this paper in nature communication.

(recommendation: not suitable for publication, resubmit or submit elsewhere)

Response: In order to demonstrate that the proposed reaction actually took place and yielded 2D-textiles consisting of interwoven polymer strands, we have disassembled the textile sheets by ultrasonication in THF. The corresponding AFM results clearly show the presence of the 1D-polymer chains.

We agree with the referee in that with the present roughness in the SURMOFs the fabrication of true single-layer textiles is too demanding. We have thus followed the advice of the referee and have changed the corresponding claim in the revised manuscript.

Reviewer #2 (Remarks to the Author):

I read this paper with joy. Attempts to create regular, large scale woven materials on a molecular scale are a major challenge and definitely worth pursuing. Achieving such goal is associated with an extraordinarily high level of static and dynamic structural control. A molecular fabric represents a breakthrough in the molecular sciences combined with a promising step ahead in terms of dimensionally restricted, new materials with a great potential for applications. I thus consider the manuscript principally important. Having said this, there are a number of deficiencies in the work presented, however, and it is thus too early for such claim to be made.

If the authors had performed the key experiment to prove a few layer or monolayer material to be a molecular fabric, evaluation of the manuscript would be easier. This experiment concerns the fact that fabrics can in principle disassemble into their constituents, which are the threads. In carpets this is prevented from happening on purpose by implementing physical knots or turning the threads after each weaving step. Because no such measures were taken by the authors, the edges of their sheets are critically important. They should suffer fraying-out phenomena that eventually result in complete disassembly. This disassembly should result in linear polymers which can be proven by conventional polymer analytical methods such as GPC.

Comparing the present manuscript with the recent publication by Yaghi in which he and his collaborators claim to have achieved 'molecular weaving' the significant step ahead is the fact that the present work provides sheet-like entities, while Yaghi's work affords 'only' microcrystalline powders. A material to choose for doing disassembly studies, is certainly a sheet and not a powder. If there happens to be kinetic hindrance, the disassembly could be enforced with AFM tips. There are useful experiments reported.

Response: We gratefully acknowledge that also this reviewer recognized the importance of the manuscript. We have followed the reviewer's suggestions to disassemble the textiles produced in our experiments into isolated single chain polymers. The disassembling was performed via a physical dispersion of the molecule weaving in THF solvent under ultrasonic condition for 6h. The disassembled single polymer chains could be clearly imaged in Fig. 5b and 5c. The height of around 4 nm for the polymer chains was received, seeing Figure 5d. We have added these results to the main text of the revised manuscript.

There are a number of issues with the manuscript some of which are delineated below.

1. What quantitative argument makes the authors believe that only the desired connections between the terminal acetylene units are formed? Given the rotational freedom of the meta-linked units – by gut feeling - other connections should be feasible as well. The possible built-up of ring strain associated with this is likely to be very small and in any case distributes over a large number of atoms

Response: We agree with the referee in that other connections aside from the one proposed might be possible. It is important to realize, however, that for such other connections a substantial, energetically very costly distortion of the framework is required (see Figure S12 in the supporting information). Since the linear polymer chains imaged in the AFM are very uniform (see Figure 5 in the revised main text), we feel we can rule out these other reaction types.

2. What makes the authors state that the acetylene conversion is approximately 90%. Looking at Figure 3b quantification is not an easy task but by visual inspection the present reviewer would estimate at a significantly lower value (possibly 60-70%; which is already an achievement). For a 'true' woven network, as the authors refer to their product, 100% conversion would be needed.

Response: The conversion yield was calculated from the integrated area of the alkyne IR band at 3295 cm^{-1} before and after the coupling reaction. When analyzing different samples, we found that the maximum yield was 89.6 %. The reviewer is correct in stating that the analysis of the results presented in figure 3b gives a yield of 78.9 %. In the new version of the manuscript we have modified the text accordingly. The referee is also correct in that in the ideal case 100% conversion is needed to convert the entire plane of the SURMOF into a single textile. This would however require that the spatial arrangement of all terminally ethynyl-functionalized subunits is perfectly controlled and harmonized over the entire plane of the SURMOF, which is not the case as the interlinking polymerization reactions are randomly distributed over the entire SURMOF plane. As a consequence of this situation, there must be situations where two ethynyls that should interlink are already fixed on opposed sides of the SURMOF plane because they belong to two fabrics that did not start with the same "periodicity". Therefore the strategy is not perfect yet and we believe that these not reacting chain ends is the reason why only flakes of fabrics are formed instead of entire planes. In addition, there is residual roughness of the SURMOFs prepared with the present set of preparation techniques and parameters which also reduces the dimensionality of the perfect SURMOF planes resulting in "open-ends" of polymer chains. We are confident that further optimization of the SURMOF process will lead to smoother films and higher conversion rates.

We feel, however, that the new AFM data (Figure 5) clearly support the main claim of our paper.

3. Another issue concerns the absence of monolayers. Obviously the authors can make layered MOFs in which sacrificial layers and active layers alternate. Why not using them as starting material for a molecular fabric? This should work if all the assumptions made actually apply.

Response: We followed the strategy suggested by this reviewer. Unfortunately, the residual roughness already referred to above prohibited the formation of larger monolayers. Again we are sure that future optimizations of the SURMOF process will allow doing such experiments. We feel, however, that the achievements documented in the revised manuscript are sufficient to warrant publication of our data.

4. On a side note: The strength of a fabric does not normally outperform a covalent network but rather its ductility. Graphene has an enormous strength.

Response: The referee is correct. We have revised the manuscript accordingly.

The reviewer feels being in sort of a dilemma. On the one hand the idea is beautiful and work in this direction should be pursued by all means. On the other hand, there are a number of serious issues (some of which are mentioned) which do not convince the reviewer about the full correctness of the claims made. If this submission was a perspective paper, it could be possibly accepted. As full scale research paper though this cannot be the recommendation. The present reviewer would be very willing to reconsider if substantial improvements have been achieved.

Response: We gratefully acknowledge that the referee states “the idea is beautiful and work in this direction should be pursued by all means”. We hope the additional AFM results obtained for the disassembled textiles together with the point-by-point rebuttals provided above will allow this specialist in the field to come to a positive recommendation. We would like to point out that our results represent the first case where a 2D textile consisting of interwoven polymer chains has been synthesized.

Reviewer #3 (Remarks to the Author):

*Review of “Molecular weaving: A bottom-up approach...” by Z. Wang et al.
The authors of this work describe a compelling strategy to use layer-by-layer MOF crystals create inter-woven single-strand polymers across large dimensions, which if realized, would constitute an interesting advance in the scaling of polymer textiles. The idea presented involves multi-hetero-layering of pillared 2D MOFs using two sets of binodal linkers, one of which couples only to the Cull metal centers, and the other with active alkyne-terminated side groups that are specifically sized to enable polymer cross-linking across the pores in the 2D polymer plane. While the research concept and description are intriguing, the primary results, including TEM and AFM analysis, unfortunately do not conclusively show that the resulting material contains interwoven molecular units as shown and described in the diagrammed scheme. Therefore, I cannot recommend publishing this article.*

Response: We are delight to learn that also the third referee clearly realizes the importance of the results described in the manuscript. We have carefully studied this reviewer's suggestions. Additional experiments were carried out, in particular we were able to disassemble the 2D-textiles and image the individual polymer chains with AFM. We are optimistic that these new results together with the point-by-point rebuttal of the referees' criticism provided below will allow him/her to come to a positive recommendation.

Specific Comments:

1. The TEM and AFM results show that the thickness of the polymer layer left behind after removing the MOF metal-centers is directly proportional to the number of layers of the MOF thin film deposited by the sequential dip method. However, this is certainly not sufficient to conclude that the molecular weaving has been achieved. In fact, this trend could also result if the molecules were deposited in randomly oriented monolayers before polymerization. To support the authors' claim, the authors should at least do control experiments varying the ABAB sequence, for example, doing AABAAB and show that the scaling trend in Figure 4 still holds. Moreover, the authors need to describe what the expected layer thickness would be for the schematic shown. Figure 4 shows 2 nm per layer which is too large for the expected laterally aligned woven molecular bilayer.

Response: We agree with this reviewer that a linear increase of the produced polymer sheets with the number of deposition sequences is not sufficient to demonstrate the proposed synthesis of interwoven, linear polymer chains. Therefore we have carried out new experiments where we have disassembled the textiles. The AFM data (see Figure 5 b and c) clearly show the presence of the polymer chains proposed on the basis of the scheme presented in Figure 5a.

2. Another concern is that for the mechanism to proceed as diagrammed, we must believe that the MOF layer forms as a uniform 2D monolayer across a large substrate surface, and that linker exchange between the layers does not occur. The similarity of the BAB-TPDC and BMB-TPDC linkers will facilitate exchange. Missing linkers are also known to often appear during MOF growth. For the process to create "perfectly interwoven polymer strands" and "perfectly preorganized...SURMOF layers" (as the authors claim) we need to assume that the process produces no linker exchange and no missing linker sites. A missing linker in non-polymerizing layer "B" for example, could be filled by a polymerizing linker during layer "A" formation. This linker intermixing will preclude formation of "true planar fabrics", in direct opposition to the authors' key claim. A careful reading of these authors' previous work on MOF hetero structures (e.g. ref#13 in current manuscript) where each hetero-layer consisted of multiple layers of each MOF, does not convincingly exclude the possibility of linker exchange or missing linkers during growth.

Response: The possibility of linker exchange when immersing a MOF built with one linker type in a solution of another linker is an important issue and we agree with the referee in that this point needs to be addressed. In fact, the same point has been raised by reviewer #1. As described in detail in the reply provided above, by determining the amount of the polymerizable linker in the SURMOFs using quantitative IR-spectroscopy, we can exclude substantial linker exchange under the experimental conditions used to grow the SURMOFs used in the present work. A discussion of this point has been added to the manuscript.

3. It is well known that in many 2D material growth strategies, such as molecular beam epitaxy, system thermodynamics can drive formation of clusters and multilayers even under conditions limited to quantitative monolayer coverage. The results showing patchy pieces after 3 cycles, and the authors' acknowledgment of the appearance of surface roughness support the view that full 2D MOF growth is not achieved.

Response: Here, the referee is very critical. What is a full 2D MOF growth? As shown in earlier papers on SURMOFs, these systems correspond to monolithic, crystalline thin film with extremely low defect (e.g. pin holes) density. Electrochemical investigations reveal that these coatings are completely blocking. A substantial electrochemical current is only seen when redox-active moieties are loaded in the SURMOFs (add reference). The MOF thin films have lateral dimensions on the order of cm, are oriented, and exhibit lateral coherence lengths of several 100 nm. We feel it is adequate to refer to such self-defined films as full 2D-MOFs.

4. *The authors also state that the MOF layering method yields “monolithic crystalline oriented MOF films.” They do not say over what lateral dimension the crystals will extend. This limitation would also determine the limit for the lateral size of the resulting “textile”. Perhaps if the authors were able to quantify the molecular weight of the resulting polymer they could compare it to the expected lateral dimension of the starting MOF crystal. It is important for the authors to describe what limits the lateral dimensions of the MOF crystals.*

Response: In following the reviewer’s suggestion, we have calculated the lateral domain size of the SURMOFs used in the present experiments using Scherrer equation to analyze the peak widths in the in-plane XRD data. The value obtained, around 80 nm, is consistent with earlier results obtained for other types of SURMOFs (Z. Wang et al., Nano Letters 2014, 14, 1526-1529). In fact, this value also nicely agrees with the length of the polymer strands observed in the AFM-data (see Figure 5 b and 5c) of around 200 nm.

5. *More information is needed related to the in-plane diffraction experiment. How thick were the films and how were they prepared for clean in-plane scattering? This information is not clearly described in the experimental section in the manuscript or SI.*

Response: We have followed the suggestion of the referee and have added more detailed information on the out-of-plane as well as on the in-plane diffraction experiments to the experimental section.

Reviewers' Comments:

Reviewer #1 (Remarks to the Author)

1. In this response, the author indeed made a very strong argument on the macroscopic precision of the layer by layer growth technique. While this is very impressive, it is still not enough to support the 'monolayer argument', which is on the molecular level. The fact that the super-lattice peak relevant to the alternating layer is not observed also indicates that the crystallinity of the SurMOF and the precision of the layer-by-layer growth is not up to the point that an argument on the molecular structure of the material can be made. I think if the authors can re-phrase the paper and weaken the claim on the molecularly precise monolayer weaving structure, the paper would be much more appealing and I think after major revision the paper could be suitable for the journal.

2. For the AFM, this data is very nice and the image is indeed unique and informative. As it is 3 nm thick, it is convincing it might be a molecular monolayer. However, the feature observed is very large (micron scale in both x and y direction), which contradicts the author's claim that these features are molecular chains. To proof the author's claim, higher resolution AFM is required. But I think this AFM data, especially the thickness of the film, is inspiring and encouraging. If the author can discuss these data more thoroughly and combine all the experimental data in a more coherent way, the paper could be accepted.

Reviewer #2 (Remarks to the Author)

While there are still many open ends, I feel that the response by the authors is adequate. In particular I am impressed that the experiment suggested actually worked. The fact that the created objects can to some degree be disassembled into features that have similarities with linear chains furthers the authors' claim substantially.

Obviously one always would like to have more but at some point things should be wrapped up. I feel that this point has come.

I am convinced that the current work will be a interest to many people and will also stimulate more research into this direction.

Science is still lightyears away from true molecular weaving and knitting but here we have a small step towards this goal.

Acceptance is recommended unconditionally.

Reviewer #3 (Remarks to the Author)

I appreciate very much the authors' consideration and response to all the referees' comments. It appears that many of the original concerns regarding linker exchange and polymerization mechanisms are now successfully addressed in the revised manuscript.

The authors have also successfully shown that material disassembly yields linear chain polymers with dimension consistent with the expected lateral dimension of the woven fabric.

A key claim in the work is the formation of planar fabrics with monomolecular precision. Concern regarding this point was raised in my original comment #2, and it was perhaps more directly noted by referee #1, comment 2. In response, the authors say they agree with referee #1 and revised the claim accordingly. However, the revised manuscript (most notably in the abstract) still states the realization of planar textiles with monomolecular thickness. This needs to be modified.

It is disappointing that the authors did not report on the suggested experiment to include multiple inert layers between each polymerizable layer. Including multiple inert layers would support the trend claimed in Figure 4j that the thickness scales only with the number of polymerized layers, and it would add further evidence to the claim for selective co-planar polymerization. Also, including multiple inert layers would increase the spacing between the polymerized layers, resulting in less interplane interaction during dissolution, and perhaps allowing partial separation of a single layer which could then be observed by AFM imaging. Did the authors consider this experiment?

Response to reviewers' comments.

First of all, we would like to thank all three reviewers for taking the time to look again at our manuscript. We are pleased that all acknowledged that the quality of the paper has been improved considerably during the major revision of our paper. While the second referee recommended unconditional acceptance, the first and the third referee had some more minor concerns. Please find below a point-by-point rebuttal of this criticism, together with a list of changes made to the manuscript.

Reviewer #1 (Remarks to the Author):

1. In this response, the author indeed made a very strong argument on the macroscopic precision of the layer by layer growth technique. While this is very impressive, it is still not enough to support the 'monolayer argument', which is on the molecular level. The fact that the super-lattice peak relevant to the alternating layer is not observed also indicates that the crystallinity of the SurMOF and the precision of the layer-by-layer growth is not up to the point that an argument on the molecular structure of the material can be made. I think if the authors can re-phrase the paper and weaken the claim on the molecularly precise monolayer weaving structure, the paper would be much more appealing and I think after major revision the paper could be suitable for the journal.

Response: We are very pleased to learn that the referee considers our response as “very strong and impressive”. We have followed his advice and subjected the manuscript to a major revision, during which we weakened the claim of precise monolayer weaving. In particular, the abstract has been modified (“A major step towards” instead of “has been reached”), and a further discussion has been added to the text (see next point).

2. For the AFM, this data is very nice and the image is indeed unique and informative. As it is 3 nm thick, it is convincing it might be a molecular monolayer. However, the feature observed is very large (micron scale in both x and y direction), which contradicts the author's claim that these features are molecular chains. To proof the author's claim, higher resolution AFM is required. But I think this AFM data, especially the thickness of the film, is inspiring and encouraging. If the author can discuss these data more thoroughly and combine all the experimental data in a more coherent way, the paper could be accepted.

Response: We gratefully acknowledge the very positive comments from this reviewer on our AFM images. As the reviewer correctly stated, the thickness of the fibers observed after

disassembling the textiles amounts to around 3 nm, which corroborates our claim of textile layers made from single molecular chains. In particular, it demonstrates that we were able to make double-layer textiles, since a thickness of around 5 nm was observed in the AFM data recorded for the fabric (see Fig. 4h.) To make this point more clear, the following text has been added to the manuscript: "The thickness of around 5 nm observed for the fabric prior to disassembly (Fig. 4h) together with the thickness of the individual polymer chains (around 4 nm, see above) suggests that the fabricated textile sheets consist of at most double layers."

Reviewer #2 (Remarks to the Author):

While there are still many open ends, I feel that the response by the authors is adequate. In particular I am impressed that the experiment suggested actually worked. The fact that the created objects can to some degree be disassembled into features that have similarities with linear chains furthers the authors' claim substantially.

Obviously one always would like to have more but at some point things should be wrapped up. I feel that this point has come.

I am convinced that the current work will be a interest to many people and will also stimulate more research into this direction.

Science is still lightyears away from true molecular weaving and knitting but here we have a small step towards this goal.

Acceptance is recommended unconditionally.

Response: We are delighted to read the positive comments from this reviewer.

Reviewer #3 (Remarks to the Author):

I appreciate very much the authors' consideration and response to all the referees' comments. It appears that many of the original concerns regarding linker exchange and polymerization mechanisms are now successfully addressed in the revised manuscript.

The authors have also successfully shown that material disassembly yields linear chain polymers with dimension consistent with the expected lateral dimension of the woven fabric.

Response: We are very pleased to learn that this reviewer considers the modifications and additional material added to the manuscript adequate.

A key claim in the work is the formation of planar fabrics with monomolecular precision. Concern regarding this point was raised in my original comment #2, and it was perhaps more directly noted by referee #1, comment 2. In response, the authors say they agree with referee #1 and revised the claim accordingly. However, the revised manuscript (most notably in the abstract) still states the realization of planar textiles with monomolecular thickness. This needs to be modified.

Response: As already mentioned in the response to reviewer #1's comment 1, we have followed the reviewer's suggestion and weakened our claim on the successful fabrication of a textile monolayer. While we believe that our method is in principle suited to achieve this ultimate goal, the present quality of the interfaces in the multilayer SURMOFs is not sufficient to realize true monolayer thickness. The text has been changed accordingly.

It is disappointing that the authors did not report on the suggested experiment to include multiple inert layers between each polymerizable layer. Including multiple inert layers would support the trend claimed in Figure 4j that the thickness scales only with the number of polymerized layers, and it would add further evidence to the claim for selective co-planar polymerization. Also, including multiple inert layers would increase the spacing between the polymerized layers, resulting in less interplane interaction during dissolution, and perhaps

allowing partial separation of a single layer which could then be observed by AFM imaging. Did the authors consider this experiment?

Response: We thank the referee for the close inspection of our data. In fact, Figure 4j already demonstrated a linear increase of the thickness with the number of polymerized layers. Since the goal of the experiments is to produce single layers of textiles consisting of interwoven polymer fibers, we do not see the relevance of the suggested experiment with multiple inert/per-polymer layers. If successful, this process would lead to stack of 2D-textiles. Furthermore, as already stated above, at present we are limited by the roughness of the interfaces between the inert SURMOFs layer and the pre-polymer SURMOF layer. Since this roughness will increase from the bottom to the top, the second, third, fourth pre-polymer layers sandwiched in between inert layer will have structural qualities (e.g. patch sizes) inferior to the first layer.

For these reasons, we doubt that these additional experiments would be constructive.